



**Study on Multi-water Sources Allocation Based on Multi-scenario potential**
**tapping under Extreme Drought: An Example from the Yellow River Water**
**Supply Area in Henan**
**Fang Wan[1,2], Shaoming Peng[3], Yu Wang[4*], Xiaokang Zheng[5], Fei Zhang[1], Weihao Wang[6],**
**Xiaohui Shen[1]**
[1]North China University of Water Resources and Electric Power, ZhengZhou 450045, China;
[2]Key Laboratory of Water Management and Water Security for Yellow River Basin, Ministry of
Water Resources(under construction)Zhengzhou 450003, China;
[3]MWR General Institute of Water Resources and Hydropower Planning and Design (GIWP),
Beijing 100032, China;
[4]Yellow River Conservancy Commission, ZhengZhou 450003, China;
[5]Yellow River Engineering Consulting Co., Ltd. ZhengZhou 450003, China;
[6]China Institute of Water Resources and Hydropower Research, BeiJing, 100038, China.
Correspondence to: Yu Wang (wanxf1023@163.com)
**Abstract:** The water supply of water resources allocation under extreme drought is insufficient,
and the limited available water resources make it urgent to tap the potential of water supply. In this
paper, the Yellow River water supply area in Henan Province is taken as an example to study the
multi-water source allocation under extreme drought. According to the Palmer Drought Severity
Index (PDSI), the extreme drought years are selected, and the water supply and demand balance in
the extreme drought years is analyzed, and the water shortage degree of each water supply area is
obtained. In this paper, unconventional water, flood resource utilization and elastic exploitation of



groundwater are used as potential water sources. Different water supply scenarios are set up
according to different potential tapping measures, and multi-scenario supply increase under
extreme drought is explored. A multi-water source allocation model with the goal of minimizing
water shortage is constructed, and a multi-scenario supply increase allocation scheme is proposed,
which provides a basis for the study of water supply increase allocation to alleviate the drought
degree of the the Yellow River Water Supply Area in Henan. Through the Multi-scenario potential
tapping of multiple water sources, the existing potential water volume can be maximized, which is
conducive to reducing the water supply pressure and water use restrictions of conventional water
sources, improving the support capacity and guarantee capacity of water resources, and reducing
the economic and social development bottlenecks caused by extreme drought.
**Keywords:** Multi-water sources allocation; Multi-scenario potential tapping; Palmer Drought
Severity Index; the Yellow River Water Supply Area in Henan
**Introduction**

Drought is a common and complex natural disaster. In the past 50 years, 34% of the people

affected by natural disasters in the world have been affected by drought (WMO, 2021). The sixth
assessment report of the Intergovernmental Panel on Climate Change (IPCC) (IPCC, 2021)
pointed out: Climate change is exacerbating the uneven water cycle. High temperatures and
multiple types of drought events are frequent in parallel, causing varying degrees of drought in
many regions. In recent years, the dry and wet imbalance in the Yellow River Water Supply Area
in Henan with monsoon climate characteristics. According to the "China Drought Disaster Data
Set" (Zhu et al, 2018), the Yellow River Basin has staged 5~6 extreme droughts in the past 50
years. The extreme drought events in the Yellow River Basin are staged every 10 years (Zhu et



al,2018; Zheng et al, 2022), which limits the development of industrial and agricultural production,
life development, ecological environment and other aspects. With the development of social
economy, the allocation of conventional water resources has been unable to meet the expected
demand. The Ministry of Water Resources "Guidance on the Integration of Unconventional Water
Resources into the Unified Allocation of Water Resources" (Zhou, 2017) emphasizes the
importance of the potential research of unconventional water resources, so that it can play a
guarantee role in the water resources allocation system, and requires the use of various means to
expand the scope of allocation and increase the proportion of water distribution. Therefore, on the
basis of the existing conventional water resources, it will be an important measure to alleviate the
drought in the Yellow River Water Supply Area in Henan by tapping the potential and increasing
the supply, and integrating unconventional water, flood resources and elastic exploitation of
groundwater into the water source system of water resources allocation.
In recent years, the theory and technology of conventional water resources allocation have
become more mature, and the research on optimal allocation of water resources has attracted
attention at home and abroad. Wang Yu et al. (2021) scientifically set the water diversion index in
the river according to the incoming water situation, and at the same time consider fairness and
efficiency, and increase the saved water supply to the provinces along the Yellow River; Yang
Mingzhi et al. (2022) regarded social water use and natural hydrology as the research object,
studied the feedback between the two processes, and developed a distributed allocation model
based on the water cycle; Tan et al. (2018) took the unilateral water benefit as the objective
function, considered the fractional programming and robust optimization at the same time, and
established the water resources optimization model, which improved the utilization efficiency of



agricultural water; Ren et al. (2017) gave full play to the advantages of multi-objective fuzzy
programming, constructed a multi-objective model of multiple benefits, rationally planned land
use and irrigation water, and obtained an effective and fair irrigation plan; Aiming at the prediction
of water supply and demand and its comprehensive value, Zhang et al. (2023) used the WRA
model to study the coordination and stable development of each system, used the emergy analysis
method to carry out quantitative analysis, reasonably calculated the base year and the planning
year, and proposed a sustainable water distribution plan; Sperotto A et al. (2019) discussed the
application of multi-scenario analysis method based on Bayesian network in water quality
sustainability assessment under uncertain conditions. Razavi S et al. (2014) used a multi-scenario,
multi-reservoir optimization method to evaluate these new control structures, and proposed an
optimization model based on dynamic programming; Marques J et al. (2015) proposed a new ROs
method to deal with the uncertainty and two conflicting objectives throughout the planning scope,
and defined some new possible expansion areas for different scenarios; Banadkooki F B et al.
(2022) discussed the optimal allocation of water resources in arid basins, analyzed 13 scenarios,
and determined the optimal solution by comparing the results of GA and NSGA-II optimization
techniques; Balla K M et al. (2020) established a multi-scenario MPC ( MS-MPC ) method to deal
with the uncertainties of the expected inflow of urban drainage network ( UDN ). Most of the
above studies use conventional water as the configuration water source. The water source
configuration is relatively simple, and the scenario selection lacks consideration of multiple water
sources. The optimization of the model and the improvement of the configuration method are
limited. At present, the utilization of rainwater and other unconventional water sources in the
Yellow River Water Supply Area in Henan is only 465 million m3 (2021 Annual "China Water





Resources Bulletin〞released, 2018). Therefore, this paper will start from the perspective of
tapping potential and increasing supply, and take unconventional water, flood resources and elastic
exploitation of groundwater as new water sources to study the allocation of water resources in the
Yellow River Water Supply Area in Henan under extreme drought.

In this paper, the water resources allocation under different potential tapping scenarios under

extreme drought is taken as the research focus, and the Palmer Drought Severity Index(PDSI) is
taken as the research index to study the drought structure distribution and water shortage in
extreme drought years. Based on the multi-scenario mining potential of multi-water sources, a
water resources allocation model with the goal of minimizing water shortage is constructed.
According to the water demand level and allocation principle, a multi-scenario allocation scheme
is proposed to provide a strong basis for future research on drought control measures.
**1 Overview of the study area**
**1.1 Watershed generalization**
The Yellow River Water Supply Area in Henan is located along the Yellow River, located in
the northern part of Henan Province (Figure 1), accounting for nearly half of the area of Henan
Province, the specific regional reference Table 1. The Yellow River Water Supply Area in Henan
contains 14 cities, some of which are not fully covered. At present, the total population of the
Yellow River water supply area is about 69.8 million, and the urbanization rate is 54% (Du, Liu,
and Hao, 2020). The effective irrigation area in the water supply area is 53.6 million mu, and the
actual irrigation area is 47.8 million mu (Du, Liu, and Hao, 2020). The total amount of water
supply in the cities of the Yellow River water supply area is reduced, and the unbalanced spatial
and temporal precipitation leads to a large water gap in the southwest region (Yao et al, 2018). The



gap between flood season and non-flood season is prominent, and the proportion of flood season
inflow is relatively large (about 60%~70%) (Fang et al, 2019), and the runoff is mostly distributed
in mountainous areas(Sun, 2021). In recent years, the total amount of water used in the Yellow
River water supply area has been increasing, domestic water and ecological water use have
increased, industrial water and agricultural water use have generally declined, and the overall
drought situation in the region has continued (Sun, 2021).

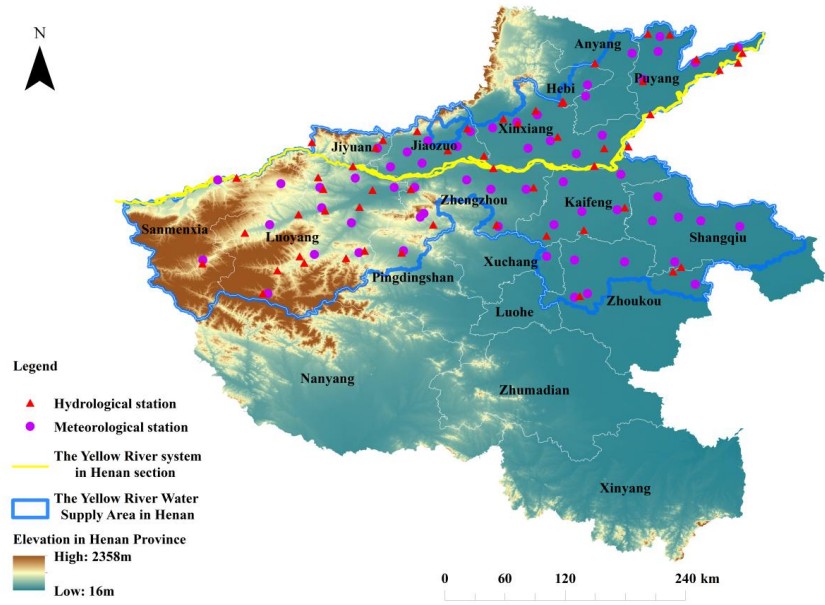


Figure 1    The administrative division of the Yellow River Water Supply Area in Henan and the

map of meteorological and hydrological stations

Table 1    The specific division scope of the Yellow River Water Supply Area in Henan

| Partition of water supply area | Specific scope | Partition | Specific scope |
|---|---|---|---|
| Zhengzhou division | Municipal district,Gongyi City,Xingyang City,Zhongmu County,Xinzheng City,Dengfeng City | Jiaozuo division | Qinyang City, Mengzhou City, Boai County, Wuzhi County, Wen County, Xiuwu County |



| Kaifeng division | Municipal districts, Qixian County, Weishi County, Lankao County, Tongxu County | Puyang division | municipal distric, Fan County, Taiqian County, Puyang County, Qingfeng County, Nanle County |
|---|---|---|---|
| Luoyang division | Municipal district, Yanshi City, Mengjin County, Yiyang County, Luoning County, Yichuan County, Xin 'an County, Luanchuan County, Song County, Ruyang County | Xuchang division | Yanling County |
| Pingdingshan division | Ruzhou City | Sanmenxia division | Municipal districts, Yima City, Mianchi County, Lushi County, Lingbao City |
| Anyang division | Hua County, Neihuang County | Shangqiu division | Municipal districts, Yucheng County, Zhecheng County, Xiayi County, Minquan County, Ningling County, Sui County |
| Hebi division | Xun County | Zhoukou division | Luyi County, Fugou County, Xihua County, Taikang County |
| Xinxiang division | Municipal distric, Weihui City, Yanjin County, Fengqiu County, Changyuan County, Xinxiang County, Huojia County, Yuanyang County | Jiyuan division | Jiyuan demonstration area |
| Total | 63 counties (cities, districts) | | |

**1.2 Basic information**
The data come from the Resource and Environmental Science and Data Center and the Henan
Provincial Water Resources Bulletin. The data of 59 meteorological stations in the Yellow River
Water Supply Area in Henan were collected. The data of precipitation, temperature and soil were
based on 58 years (1961-2018) monthly data.
**2 Selection of extreme drought years and setting of potential tapping scenarios**
**2.1 Selection of extreme drought years**
In this paper, the extreme drought years the Yellow River Water Supply Area in Henan are
selected according to the Palmer Drought Severity Index (PDSI). Figure 2 is the PDSI histogram
from 1961 to 2018, which reflects the drought change of the PDSI annual sequence in the Yellow
River Water Supply Area in Henan. According to the size of the PDSI value, the drought grade is



divided (Lu et al, 2022), as shown in Table 2.

Table 2    PDSI drought classification standard table

| PDSI | Drought level | PDSI | Drought level |
|---|---|---|---|
| (-1,1) | Normal | (-4,-3] | Serious drought |
| (-2,-1] | Light drought | (-∞,-4] | Extreme drought |
| (-3,-2] | Moderate drought | | |

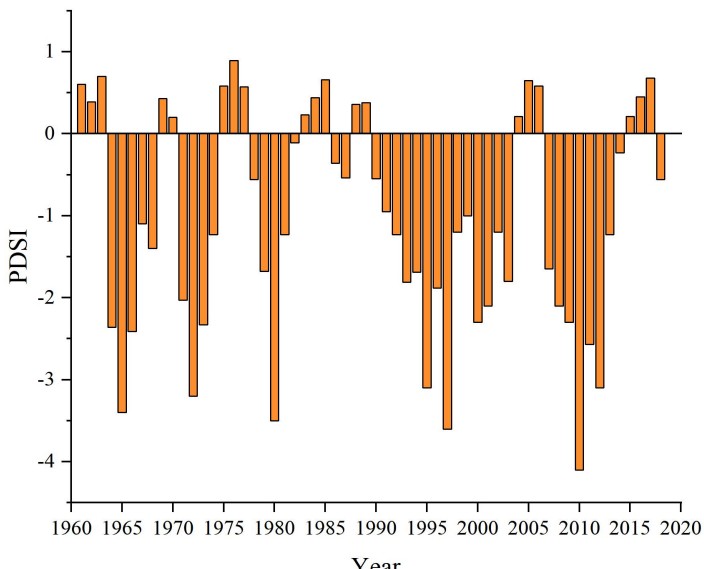

Figure 2    The overall PDSI annual sequence histogram of

the Yellow River Water Supply Area in Henan

From the above Figure 2, it can be seen that from 1961 to 2018, there were many frequent

drought years in the Yellow River Water Supply Area in Henan as a whole. Among them, serious
droughts occurred in 1965, 1972, 1980, 1995, 1997, 2010 and 2012, and serious droughts were
staged every 10 or so. The degree of drought in 2010 reached the level of extreme drought, and the
other years were mild drought or normal. Therefore, 2010 was selected as the extreme drought


year. The distribution of drought grade in each district of the Yellow River Water Supply Area in
Henan in 2010 is shown in Figure 3.

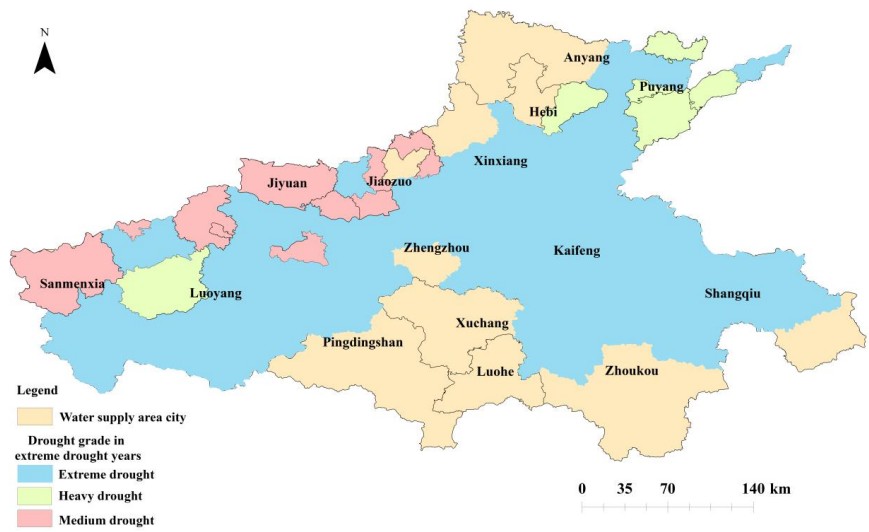

Figure 3    Distribution of drought grade in the Yellow River Water Supply Area in Henan in 2010

It can be seen from Figure3 that, on the whole, the drought degree of the Yellow River Water

Supply Area in Henan reached the level of extreme drought in 2010.Extreme drought occurred in
48 of the 63 districts and counties, and severe drought or moderate drought occurred in the
remaining districts and counties (some districts and counties in Sanmenxia City, Jiyuan City,
Jiaozuo City, Luoyang City, Puyang City and Hebi City).

According to the water resources bulletin of Henan Province in 2010 and the "Water Quota"

(Qian, 2019), the supply and demand balance of the Yellow River Water Supply Area in Henan in
2010 was analyzed, as shown in Table 3.

Table 3    The Yellow River Water Supply Area in Henan in 2010 supply and

demand balance analysis table        Unit: hundred million m³

| Partition of | Water demand in 2010 | Water supply in 2010 | Differentials | Water |
|---|---|---|---|---|


| water supply area | Agriculture | Industry | Life | Ecology | Subtotal | Surface water | Ground water | Other water | Subtotal | | shortage rate (%) |
|---|---|---|---|---|---|---|---|---|---|---|---|
| Zhengzhou division | 3.49 | 4.78 | 4.69 | 4.12 | 17.07 | 4.72 | 6.92 | 0.24 | 11.88 | -5.19 | 30.40 |
| Kaifeng division | 10.28 | 2.45 | 1.77 | 3.25 | 17.74 | 3.03 | 8.56 | 0 | 11.59 | -6.15 | 34.67 |
| Luoyang division | 4.29 | 5.84 | 2.61 | 1.84 | 14.58 | 6.19 | 5.53 | 0 | 11.72 | -2.86 | 19.61 |
| Pingdingshan division | 1.18 | 1.55 | 0.39 | 0.10 | 3.22 | 0.83 | 1.22 | 0 | 2.04 | -1.18 | 36.66 |
| Anyang division | 7.10 | 1.21 | 0.87 | 1.39 | 10.56 | 1.75 | 4.65 | 0 | 6.40 | -4.16 | 39.38 |
| Hebi division | 1.50 | 0.35 | 0.40 | 0.35 | 2.61 | 0.48 | 1.35 | 0 | 1.83 | -0.78 | 29.87 |
| Puyang division | 11.94 | 3.06 | 1.28 | 1.05 | 17.33 | 7.54 | 4.95 | 0.30 | 12.79 | -4.54 | 26.18 |
| Xuchang division | 1.04 | 0.49 | 0.28 | 0.22 | 2.03 | 0.37 | 0.65 | 0 | 1.02 | -1.01 | 49.81 |
| Sanmenxia division | 1.31 | 1.50 | 0.86 | 0.56 | 4.23 | 2.17 | 1.37 | 0.03 | 3.58 | -0.65 | 15.33 |
| Shangqiu division | 10.55 | 1.67 | 1.96 | 0.65 | 14.83 | 1.41 | 7.41 | 0 | 8.82 | -6.01 | 40.52 |
| Zhoukou division | 8.11 | 1.52 | 1.65 | 0.42 | 11.71 | 0.45 | 7.61 | 0 | 8.06 | -3.65 | 31.18 |
| Jiyuan division | 0.99 | 0.77 | 0.22 | 0.46 | 2.44 | 1.09 | 0.80 | 0.09 | 1.99 | -0.45 | 18.34 |
| Jiaozuo division | 6.54 | 3.02 | 1.00 | 0.50 | 11.07 | 3.71 | 5.17 | 0 | 8.89 | -2.18 | 19.66 |
| Xinxiang division | 12.25 | 2.04 | 1.77 | 1.16 | 17.21 | 4.85 | 5.61 | 0 | 10.46 | -6.75 | 39.24 |
| Total | 80.56 | 30.24 | 19.74 | 16.08 | 146.62 | 38.60 | 61.81 | 0.66 | 101.07 | -45.55 | 31.07 |

It can be seen from Table 3 that the total water demand the Yellow River Water Supply Area
in Henan is 14.662 billion m³, the total water supply is 10.107 billion m³, the total water shortage
is 4.555 billion m³, and the water shortage rate is 31.07%. In general, the supply and demand of
the Yellow River Water Supply Area in Henan is unbalanced in extreme drought years, and the
water shortage rate of 8 water supply areas exceeds 30%. In order to ensure the normal needs of
residents and the good ecological environment, it is urgent to tap the water supply potential of
unconventional water sources, flood water sources and elastic exploitation of groundwater.
**2.2 Potential tapping scenario setting**



162 On the basis of the structure of the original water supply source, the water source system is

163 expanded and managed according to local conditions to maximize the supply capacity of water

164 resources, such as expanding the construction of reclaimed water and rainwater harvesting

165 projects, the appropriate adjustment of the flood limit water level of the reservoir, and improving

166 the utilization rate of reclaimed water and sewage. The Multi-water sources allocation of different

167 potential tapping scenarios is to study the potential of unconventional water sources, flood water

168 resources and elastic exploitation of groundwater under extreme drought. According to different

169 supply increase measures, it is divided into three water supply scenarios to study the

170 multi-scenario supply increase under extreme drought, so as to realize the sustainability of water

171 resources supply and the maintenance of ecological environment. The development potential of

172 unconventional water sources is great. At present, the utilization form of unconventional water

173 sources in the Yellow River Water Supply Area in Henan is mainly rainwater harvesting. Under

174 extreme drought conditions, it is necessary to increase the scale of rainwater harvesting facilities,

175 improve the utilization efficiency of reclaimed water and sewage, and increase the utilization of

176 mine water; In order to improve the accuracy of flood forecasting, it is necessary to increase the

177 dam gate, make full use of natural depressions and artificial lakes, improve the water storage

178 capacity of the water storage project, speed up the storage and set the scheduling rules

179 scientifically, which is suitable for improving the normal water level of the reservoir, and the

180 potential of increasing the supply of flood resources will be greatly improved; The elastic mining

181 of groundwater is based on the unit water yield of the water source well and the spring flow to

182 carry out water-rich zoning(Zhao and Zhao, 2014).As shown in Table 4, according to the strength

183 of water-richness, different scenarios are tapped. According to the potential of supply increase





184 from small to large, it is divided into supply increase scenario 1, supply increase scenario 2 and

185 supply increase scenario 3 in turn. Different supply increase scenarios correspond to different

186 supply increase measures, as shown in Table 5.

187 Table 4 Division table of groundwater water abundance in

188 the Yellow River Water Supply Area in Henan

| Regionalization basis | Partition | | | |
|---|---|---|---|---|
| | Weak water-rich area | Medium water-rich area | Strong water-rich area | Extremely strong water-rich area |
| Unit output of water source well(m³/h·m) | $q<1$ | $1\le q<5$ | $5\le q<10$ | $q>10$ |
| Flow capacity of spring (L/s) | $Q<1$ | $1\le Q<10$ | $10\le Q<50$ | $Q<50$ |

189 Table 5 Measures for increasing supply of different potential water sources

190 under different supply scenarios

| Additional supply scenario | Unconventional Water Tapping | Flood resource utilization | Elastic groundwater exploitation |
|---|---|---|---|
| Scenario 1 | Increase the scale of rainwater harvesting facilities by 5% | Increase the scale of water storage project by 5% | Mining 15% of water source in strong water-rich area and extremely strong water-rich area |
| Scenario 2 | Expand the scale of 5% rainwater harvesting facilities; reclaimed water and sewage utilization efficiency increased by 10% | Increase the scale of water storage project by 5%; reasonable setting to speed up the recovery scheduling rules | Mining 15% of the water source in the strong and extremely rich water area; mining and excavating 10% of the water source in the medium water-rich area |
| Scenario 3 | Expand the scale of rainwater harvesting facilities by 5%;the utilization efficiency of reclaimed water and sewage is increased by 10%;increase the utilization ratio of mine water by 20% | Increase the scale of water storage project by 5%; reasonably set up the scheduling rules for accelerating the recovery of savings; dynamic adjustment of reservoir flood control level | Mining 15% of the water source in the strong and extremely rich water area; mining and excavating 10% of the water source in the medium-rich water area; mining 5% water source in weak water-rich area |

191 **3 Study on Multi-water sources allocation of Multi-scenario potential tapping**

192 **3.1 Water source analysis of Multi-scenario potential tapping**





193  Based on the potential unconventional water volume, flood resource water volume and

194  groundwater volume, Multi-scenario potential tapping is carried out, In 2010, the potential water

195  volume of unconventional water in the Yellow River Water Supply Area in Henan was 5.045

196  billion $m^3$, the potential water volume of flood resources was 10.223 billion $m^3$, and the potential

197  water volume of groundwater elastic exploitation was 9.660 billion $m^3$.According to the

198  proportion of different potential water volume and potential total water volume, it is divided into

199  three different potential water source scenarios. The potential water volume and the comparison

200  before and after potential tapping are shown in Table 6.

201  Table 6  The amount of potential tapping water in different scenarios and the comparison

202  table before and after potential tapping  Unit: hundred million $m^3$

| Water supply scenario | | Unconventional water | | Flood resource utilization | | Elastic groundwater exploitation | | Total |
|---|---|---|---|---|---|---|---|---|
| | | Quantity of water | Percentage of total potential unconventional water | Quantity of water | Percentage of total potential unconventional water | Quantity of water | Percentage of total potential unconventional water | |
| Before digging potential | | 0.39 | 0.01% | 0 | 0 | 0 | 0 | 0.39 |
| After digging potential | Scenario 1 | 5.05 | 10% | 5.11 | 5% | 4.83 | 5% | 14.99 |
| | Scenario 1 | 7.57 | 15% | 10.22 | 10% | 9.66 | 10% | 27.45 |
| | Scenario 1 | 10.09 | 20% | 15.33 | 15% | 14.49 | 15% | 39.91 |

203  According to the Table 6, there is a great potential for the development of unconventional

204  water and flood water resources. The potential water of unconventional water sources in scenarios

205  1, 2 and 3 is 505 million $m^3$, 757 million $m^3$ and 109 million $m^3$; The amount of potential water in

206  flood resource tapping scenarios 1, 2 and 3 is 511 million $m^3$, 1022 million $m^3$, and 1.533 billion

207  $m^3$. With the increase of excavation potential, the total water volume of water supply scenarios 1,

208  2 and 3 is 1.499 billion $m^3$, 2.745 billion $m^3$ and 3.991 billion $m^3$. Compared with before tapping

209  the potential, the available water supply has been greatly improved. The rational development and




utilization of unconventional water sources and flood water sources plays an important role in the
sustainable development of water resources in the future.

**3.2 Water demand level division and configuration principle**

Western sociologist Maslow's hierarchy of needs theory believes that human needs are
divided into five levels like a ladder, namely: physiological needs, security needs, social needs,
respect needs, and self-realization needs. The more these needs are met, the stronger the sense of
well-being will be. Specific to the river basin, combined with the water demand characteristics of
various departments and development conditions in different regions, the life, industry, agriculture,
and ecological water demand processes are divided into three levels, namely rigid demand, rigid
elastic demand, and elastic demand, as shown in Figure 4.

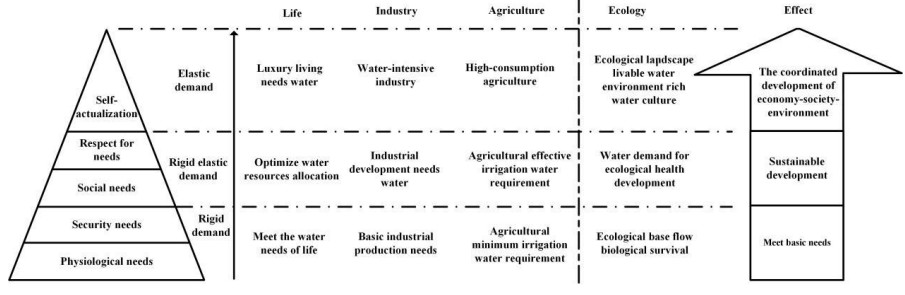

Figure 4    Water demand level division diagram

The satisfaction of each water sector is the relationship between water users according to
their water demand status and water distribution. It is generally a subjective evaluation. It is the
happiness of water users after their needs are met. When the water demand of the water sector is
met. The more, the higher the satisfaction. The runoff volume and wet and dry years are different,
and the water demand level is different. The rigid water demand is the first priority in the
distribution water, and it will be difficult to recover the loss once it is destroyed. The rigid elastic
water demand is in the second priority in the water distribution, and the loss caused by water



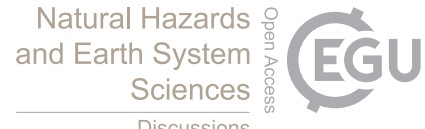
shortage is recoverable; elastic water demand is the last consideration in water resources
allocation. When the water distribution amount obtained by the water use department is at
different water demand levels, the level of satisfaction is related to different water use departments.
For example, in dry years and when the runoff is small, the water use department meets the basic
needs. At this time, the satisfaction of the agricultural department is higher than that of the
domestic, industrial and ecological water use departments.

The configuration of Multi-water sources allocation it he Yellow River Water Supply Area in

Henan should follow the following principles:

(1) The principle of matching the type of potential water sources with the water use

department. Under extreme drought conditions, groundwater sources are flexibly exploited for
domestic water and some industrial water; unconventional water sources are mainly used for
industrial water and landscape greening water; flood water resources are used for agricultural
irrigation and ecological water use.

(2) Priority principle. Priority should be given to the protection of basic domestic water.

Because the drought in the study area is mainly reflected in agriculture, agricultural water is put in
the second place, and on this basis, industrial water and ecological water are considered in turn. At
the same time, give priority to meet the rigid demand, and then ensure the rigid elastic water, and
finally meet the elastic demand.

(3) The principle of fairness. Equal distribution is needed between individual water users and

water users, and between partitions and regions in the Yellow River Water Supply Area in Henan.
Each district cooperates with each other and cooperates with the overall situation, and objectively
has the same status to allocate water sources.



(4) The principle of total control. The development of groundwater elastic mining,
unconventional water and flood water resources in the Yellow River Water Supply Area in Henan
should be matched with the carrying capacity of local water resources, reasonably control the
polluting environmental industries and high-intensity water consumption industries, and
strengthen the supervision of their total amount.
(5) The principle of sustainable use. In the case of widespread drought and complex water
shortage structure in the Yellow River Water Supply Area in Henan, a fair, effective, reasonable
and scientific setting of the water distribution ratio between life, agriculture, industry and ecology
is conducive to the benign maintenance of water resources regeneration mechanism and
sustainability.
**3.3 Model construction**
(1) Taking the minimum water shortage as the objective function:
$$\min Z_p = \sum_{c}^{h} \sum_{d}^{l} (W_{c,p} - W_{d,p}) \tag{1}$$

Where $Z_p$ is the total water deficit in the scenario of enemy $p$, hundred million m³, $p =$
(1,2,3); $W_c$ is the water supply of type $i$ users in scenario $p$, hundred million m³, $c =$
(1,2,3… $h$); $W_j$ is the water demand of type $d$ user in type $p$ scenario, hundred million
m³, $d =$(1,2,3… $l$).
(2) Constraint conditions
Constraints of available water supply and water demand:
$$\sum_{i}^{m} W_i \leq W_G \tag{2}$$

$$\sum_{j}^{n} W_j \leq W_X \tag{3}$$





$$W_G = W_B + W_D + W_C + W_H \tag{4}$$

$$W_X = W_S + W_Y + W_N + W_T \tag{5}$$

In the formula, $W_G$ is the total amount of water available, billion m³; $W_B$ is the available
water supply of surface water, billion m³; $W_D$ is the available water supply of groundwater,
billion m³; $W_X$ is the total water demand, billion m³; $W_S$ is the total amount of domestic water
demand, billion m³; $W_Y$ is the total amount of industrial water demand, billion m³; $W_N$ is the
total amount of agricultural water demand, billion m³; $W_T$ is the total ecological water demand,
billion m³.
Unconventional water and flood water resources constraints:
$$\sum_r^a W_{C_{p,r}} \leq W_{C_0} \tag{6}$$

$$\sum_k^b W_{H_{p,k}} \leq W_{H_0} \tag{7}$$

$$\frac{\sum_r^a W_{C_{p,r}} + \sum_k^b W_{H_{p,k}}}{W_G} \geq \alpha \tag{8}$$

In the formula, $W_{C_{p,r}}$ is the unconventional water supply of type $r$ users in scenario $p$,
billion m³, $r = (1,2,3... a$ ); $W_{C_0}$ is the original unconventional water supply, billion m³ ;
$W_{H_{p,k}}$ is the amount of water supply for flood water resources of the type $k$ user in the $p$ th
scenario, 100 million m³, $k = (1,2,3... b$ ); $W_{H_0}$ for the original flood water resources, water
supply, billion m³ ; $\alpha$ is the reasonable proportion coefficient of unconventional water and flood
water resources.
Priority constraint. The order of water supply is as follows:
$$W_1 > W_2 > W_3 > W_4 \tag{9}$$


$$W_{\mathrm{I}} > W_{\mathrm{II}} > W_{\mathrm{III}}$$
(10)

$W_1$ is the order of domestic water supply; $W_2$ the order of supply for agriculture; $W_3$ is the
order of industrial water supply; $W_4$ is the order of ecological water supply; $W_{\mathrm{I}}$ is the order of
rigid water demand; $W_{\mathrm{II}}$ is the order of rigid elastic water demand; $W_{\mathrm{III}}$ is the order of elastic
water demand.
Groundwater depth constraint:

$$D_e \geq D_f$$
(11)

In the formula, $D_0$ is the average groundwater depth of the e region, m; $D_f$ is the maximu
m allowable groundwater depth in zone f, m.
The minimum ecological flow constraint is:
$$E_{g,t} < H_{g,t}$$
(12)

In the formula, $E_{g,t}$ and $H_{g,t}$ are the minimum ecological flow and the actual flow in the t
period of the g ecological control section, m³/s.
Non-negative constraints: All variables are greater than or equal to 0.
**4 Scenario configuration scheme analysis**
**4.1 Scheme calculation**
According to the distribution of water shortage and drought grade in each area of the Yellow
River Water Supply Area in Henan in extreme drought years, the model is solved according to the
configuration principle and water use constraint conditions. The solution process is based on three
different water supply scenarios to obtain three schemes for increasing supply. The configuration
results are shown in Table 7



Table 7    Results table of water resources allocation in the Yellow River Water Supply Area in

Henan    Unit: hundred million m³

| Type of water supply | Life | | | Industry | | | Ecology | | |
| --- | --- | --- | --- | --- | --- | --- | --- | --- | --- |
| Partition of water supply area | Scenario 1 | Scenario 2 | Scenario 3 | Scenario 1 | Scenario 2 | Scenario 3 | Scenario 1 | Scenario 2 | Scenario 3 |
| Zhengzhou division | 4.69 | 4.69 | 4.69 | 3.34 | 3.88 | 4.45 | 2.56 | 3.11 | 3.61 |
| Kaifeng division | 1.77 | 1.77 | 1.77 | 1.71 | 1.99 | 2.28 | 2.02 | 2.45 | 2.85 |
| Luoyang division | 2.61 | 2.61 | 2.61 | 4.08 | 4.74 | 5.43 | 1.14 | 1.39 | 1.62 |
| Pingdingshan division | 0.39 | 0.39 | 0.39 | 1.08 | 1.26 | 1.44 | 0.06 | 0.08 | 0.09 |
| Anyang division | 0.87 | 0.87 | 0.87 | 0.84 | 0.98 | 1.12 | 0.86 | 1.05 | 1.22 |
| Hebi division | 0.4 | 0.4 | 0.4 | 0.25 | 0.29 | 0.33 | 0.22 | 0.27 | 0.31 |
| Puyang division | 1.28 | 1.28 | 1.28 | 2.14 | 2.48 | 2.84 | 0.65 | 0.79 | 0.92 |
| Xuchang division | 0.28 | 0.28 | 0.28 | 0.34 | 0.4 | 0.46 | 0.14 | 0.17 | 0.2 |
| Sanmenxia division | 0.86 | 0.86 | 0.86 | 1.05 | 1.22 | 1.39 | 0.35 | 0.42 | 0.49 |
| Shangqiu division | 1.96 | 1.96 | 1.96 | 1.17 | 1.36 | 1.55 | 0.41 | 0.49 | 0.57 |
| Zhoukou division | 1.65 | 1.65 | 1.65 | 1.07 | 1.24 | 1.42 | 0.26 | 0.32 | 0.37 |
| Jiyuan division | 0.22 | 0.22 | 0.22 | 0.54 | 0.62 | 0.71 | 0.28 | 0.34 | 0.4 |
| Jiaozuo division | 1 | 1 | 1 | 2.11 | 2.45 | 2.81 | 0.31 | 0.38 | 0.44 |
| Xinxiang division | 1.77 | 1.77 | 1.77 | 1.43 | 1.66 | 1.9 | 0.72 | 0.88 | 1.02 |
| Total | 19.74 | 19.74 | 19.74 | 21.14 | 24.56 | 28.14 | 9.99 | 12.13 | 14.11 |

Table 7 Schedule




| Type of water supply | Agriculture | | | Subtotal | | | Water deficit | | |
|---|---|---|---|---|---|---|---|---|---|
| Partition of water supply area | Scenario 1 | Scenario 2 | Scenario 3 | Scenario 1 | Scenario 2 | Scenario 3 | Scenario 1 | Scenario 2 | Scenario 3 |
| Zhengzhou division | 2.82 | 3.12 | 3.42 | 13.41 | 14.8 | 16.17 | 3.66 | 2.27 | 0.9 |
| Kaifeng division | 8.32 | 9.2 | 10.08 | 13.82 | 15.41 | 16.98 | 3.92 | 2.33 | 0.76 |
| Luoyang division | 3.47 | 3.84 | 4.21 | 11.31 | 12.58 | 13.87 | 3.27 | 2 | 0.71 |
| Pingdingshan division | 0.95 | 1.06 | 1.16 | 2.49 | 2.78 | 3.08 | 0.73 | 0.44 | 0.14 |
| Anyang division | 5.74 | 6.35 | 6.96 | 8.32 | 9.25 | 10.17 | 2.24 | 1.31 | 0.39 |
| Hebi division | 1.21 | 1.34 | 1.47 | 2.08 | 2.29 | 2.51 | 0.53 | 0.32 | 0.1 |
| Puyang division | 9.66 | 10.68 | 11.71 | 13.73 | 15.24 | 16.75 | 3.6 | 2.09 | 0.57 |
| Xuchang division | 0.84 | 0.93 | 1.02 | 1.61 | 1.78 | 1.96 | 0.43 | 0.25 | 0.08 |
| Sanmenxia division | 1.06 | 1.17 | 1.28 | 3.31 | 3.67 | 4.03 | 0.91 | 0.56 | 0.2 |
| Shangqiu division | 8.54 | 9.44 | 10.35 | 12.07 | 13.25 | 14.43 | 2.76 | 1.58 | 0.39 |
| Zhoukou division | 6.57 | 7.26 | 7.96 | 9.54 | 10.47 | 11.39 | 2.17 | 1.24 | 0.32 |
| Jiyuan division | 0.8 | 0.89 | 0.97 | 1.84 | 2.07 | 2.3 | 0.6 | 0.36 | 0.13 |
| Jiaozuo division | 5.3 | 5.86 | 6.42 | 8.72 | 9.69 | 10.67 | 2.34 | 1.38 | 0.4 |
| Xinxiang division | 9.91 | 10.96 | 12.01 | 13.83 | 15.27 | 16.7 | 3.38 | 1.95 | 0.52 |
| Total | 65.21 | 72.11 | 79.01 | 116.08 | 128.54 | 141 | 30.53 | 18.07 | 5.61 |

316  According to Table 7, based on different potential tapping levels, the configuration schemes

317  of different scenarios are obtained. With the increase of the proportion of water supply in the

318  potential tapping water, the water shortage in each partition is significantly reduced, and the water

319  supply and demand are close to balance. Under different water supply scenarios, the total water



supply in the Yellow River Water Supply Area in Henan increased from 11.07 billion m$^3$ to 11.608
billion m$^3$, 12.854 billion m$^3$, and 14.100 billion m$^3$, respectively, and the total water shortage
decreased from 4.555 billion m$^3$ to 3.053 billion m$^3$, 18.07 billion m$^3$, and 5.61 billion m$^3$.
**4.2 Discussion of the scheme**
(1) Comparative analysis of tapping potential water
According to the water resources bulletin of Henan Province in 2010, in addition to
conventional water, the water supply volume of unconventional water in Henan Province in 2010
was 39 million m$^3$. In this paper, through the excavation of the potential water volume of
unconventional water, the potential water volume of flood resources and the elastic exploitation of
groundwater in 2010, the total water volume of potential exploitation based on different potential
exploitation scenarios was 1.499 billion m$^3$, 2.745 billion m$^3$ and 3.991 billion m$^3$, respectively.
Therefore, the effect of tapping potential and increasing supply in the Yellow River Water Supply
Area in Henan is obvious.
(2) Comparative analysis of department water supply
The water supply data of the water resources bulletin department of Henan Province in 2010
are compared with the results of the three scenarios after tapping the potential, as shown in Table 8.
After the allocation, compared with the water resources bulletin of Henan Province in 2010, the
water supply under different scenarios increased, which alleviated the water pressure of various
departments and realized the improvement of social and economic benefits and the benign
development of ecological environment.
Table 8    Comparative analysis table of department water supply after tapping potential
Unit: hundred million m$^3$




| Department | | Life | Industry | Zoology | Agriculture | Total water supply quantity | Total water requirement |
|---|---|---|---|---|---|---|---|
| Henan Water Resources Bulletin 2010 | | 19.74 | 19.38 | 8.98 | 52.97 | 101.07 | |
| After digging potential | Scenario 1 | 19.74 | 21.14 | 9.99 | 65.21 | 116.08 | 146.62 |
| | Scenario 2 | 19.74 | 24.56 | 12.13 | 72.11 | 128.54 | 146.62 |
| | Scenario 3 | 19.74 | 28.14 | 14.11 | 79.01 | 141.00 | 146.62 |

(3) Mitigation analysis of water shortage
The comparative analysis of the water shortage results of each water supply scenario is
shown in Table 9. On the whole, the water shortage situation in the Yellow River Water Supply
Area in Henan has been significantly improved compared with that before tapping the potential.
Under different supply scenarios, the average water shortage in the water supply area decreased
from 325 million $m^3$ to 218 million $m^3$, 129 million $m^3$ and 40 million $m^3$ respectively. The overall
water shortage rate decreased from 31.07% to 20.83%, 12.33% and 3.83%, respectively. Among
them, there is no water shortage in domestic water. Through the potential tapping of multiple
scenarios, the water shortage rate of industrial water decreased from 35.90% to 30.09%, 18.78%
and 6.94% respectively. The water shortage rate of ecological water decreased from 44.12% to
37.87%, 24.56% and 12.25%, respectively; The water shortage rate of agricultural water decreased
from 34.25% to 19.05%, 10.49% and 1.92% respectively. After Multi-scenario potential tapping,
the water shortage rate of each department (except domestic water) has been greatly reduced, but
there is still a shortage of water in industrial water, ecological water and agricultural water. In the
future, it is urgent to deepen the excavation of unconventional water sources and flood water
sources, so that the Yellow River Water Supply Area in Henan has sustainable water resources
support.
Table 9    Comparison table of water shortage results before and after tapping the potential of

the Yellow River Water Supply Area in Henan          Unit: hundred million $m^3$





| Analysis index | | Domestic water shortage rate (%) | Industrial water shortage rate (%) | Ecological water shortage rate (%) | Agricultural water shortage rate (%) | Mean value of water shortage in each district | Water shortage rate (%) |
|---|---|---|---|---|---|---|---|
| Before digging potential | | 0 | 35.90 | 44.12 | 34.25 | 3.25 | 31.07 |
| After digging potential | Scenario 1 | 0 | 30.09 | 37.87 | 19.05 | 2.18 | 20.83 |
| | Scenario 2 | 0 | 18.78 | 24.56 | 10.49 | 1.29 | 12.33 |
| | Scenario 3 | 0 | 6.94 | 12.25 | 1.92 | 0.40 | 3.83 |

(4) Configuration effect of multi-level water demand
Based on three water demand levels (rigid demand, rigid elastic demand and elastic demand),
this paper explores the potential of multiple water sources. Different water supply scenarios
correspond to different water demand levels. Water supply scenarios 1, 2 and 3 correspond to rigid
demand, rigid elastic demand and elastic demand respectively, and a multi-level water demand
configuration scheme is obtained. Table 10 shows the water demand satisfaction under different
water demand conditions. Through the increase of different potential tapping scenarios, the overall
satisfaction rates of rigid demand, rigid elastic demand and elastic demand have reached 79.17,
87.67 and 96.17 respectively. Specifically, under the condition of rigid water demand, the
satisfaction rates of domestic water demand, industrial water demand, ecological water demand
and agricultural water demand are 100%, 69.91%, 62.13% and 80.95% respectively. Under the
condition of rigid elastic demand, the satisfaction rates of domestic water demand, industrial water
demand, ecological water demand and agricultural water demand are 100%, 81.22%, 75.44% and
89.51% respectively. Under the condition of elastic demand, the satisfaction rates of domestic
water demand, industrial water demand, ecological water demand and agricultural water demand
are 100%, 93.06%, 87.75% and 98.08% respectively.Based on different potential tapping
scenarios corresponding to different water demand levels, in the future, water demand can be
considered to be hierarchical to achieve precise potential tapping and supply increase and





scientific configuration.
Table 10 is the unit of water demand satisfaction under different water demand conditions: %

| Analysis index | | Domestic water demand satisfaction rate | Industrial water demand satisfaction rate | Ecological water demand satisfaction rate | Agricultural water demand satisfaction rate | The average water demand satisfaction rate of each district |
|---|---|---|---|---|---|---|
| Water requirement levels | Rigid demand | 100 | 69.91 | 62.13 | 80.95 | 79.17 |
| | Rigid elastic demand | 100 | 81.22 | 75.44 | 89.51 | 87.67 |
| | Elastic demand | 100 | 93.06 | 87.75 | 98.08 | 96.17 |

**4.3 Discussion**
In terms of research results, the total water demand in extreme drought years in this paper is
146.62 billion m3, and the total water supply in water supply scenarios 1,2 and 3 is 116.08 billion
m3,128.54 billion m3 and 141.00 billion m3, respectively. Similar to the research results of other
scholars,(Sun X, 2021: Wang X, 2024) due to the different spatial scales, water supply structures
and methods of research, the results are different.
In terms of research methods, this paper constructs a model with the minimum water shortage
as the goal, and uses genetic algorithm and dynamic programming to calculate the configuration
model. The limited objective function and constraint conditions in the multi-water source
configuration model are difficult to fully consider the current situation and development
requirements of the water supply area. In the future, it is necessary to improve the applicability of
multi-water source users and configuration models, and at the same time, it is necessary to
comprehensively consider high-quality development goals such as economic growth, water supply



safety, and ecological health.
In terms of research perspective, this paper takes the multi-water source potential tapping and
multi-scenario supply increase configuration in extreme drought years as the research highlights.
The water supply sources mainly include unconventional water, flood resource water source and
groundwater elastic mining water source. Through the potential tapping of different measures, the
drought situation in the water supply area has been greatly alleviated. However, there are water
supply safety problems in the exploitation of flood resource water and groundwater, especially the
realization of flood resource through the regulation of reservoir flood limit water level, which
makes the contradiction between benefit and flood control safety. In the future, we can consider
vigorously exploiting unconventional water sources to achieve the reduction of water supply risk
and the benign maintenance of ecological environment.
**5 Conclusion**
Aiming at the problem of multi-resource allocation under extreme drought the Yellow River
Water Supply Area in Henan, this paper sets different water supply scenarios through different
potential tapping measures, constructs a Multi-water sources allocation model based on multiple
scenarios, and proposes three schemes for increasing supply. The main conclusions of this paper
are as follows: (1) Under extreme drought, the amount of potential tapping water has been
significantly improved through different scenarios of potential tapping measures. The amount of
potential tapping water in the three scenarios is 1.499 billion $m^3$, 2.745 billion $m^3$ and 3.991
billion $m^3$, respectively. The potential of unconventional water and flood resources the Yellow
River Water Supply Area in Henan is great. The results of potential tapping provide practical value
for the study of reasonable potential tapping of unconventional water sources. (2) After the



416 allocation, the overall water shortage degree of the Yellow River Water Supply Area in Henan has

417 been significantly alleviated. Under different water supply scenarios, the total water shortage has

418 been reduced from 4.555 billion m³ to 3.054 billion m³,1.808 billion m³ and 5.62 billion m³,

419 respectively. The overall water shortage rate decreased from 31.07% to 20.83%, 12.33% and

420 3.83%. (3) At present, the potential of groundwater is very limited, and the potential space of

421 unconventional water source and flood resource water source is very large. Under different

422 scenarios, the sum of the potential water volume of unconventional water source and flood

423 resource water source and the total water supply before the potential is more than 10%. Under

424 extreme drought conditions, this paper studies the water resources allocation of Multi-scenario

425 potential tapping and increasing supply. In the actual water distribution, the water quality of the

426 water distribution is subject to various complex requirements and restrictions, such as different

427 water users have different water quality requirements. In the future research, the joint allocation of

428 water quantity and water quality can be further considered.

429 **Declarations:** We declare that we have no conflict of interest or the authors do not have any

430 possible conflicts of interest, the authors are not affiliated with or involved with any organisation

431 or entity with any financial interest or non-financial interest in the subject matter or materials

432 discussed in this paper.

433 **Funding:** The research was supported by the National Key Research and Development Program

434 of China (2022YFC3202300), Major Science and Technology Special Projects in Henan Province

435 (201300311400), Key Laboratory of Water Management and Water Security for Yellow

436 River Basin, Ministry of Water Resources(under construction)(2023-SYSJJ-05), Open

437 Fund of Key Laboratory of Flood & Drought Disaster Defense, the Ministry of Water

438 Resources(KYFB202307260036), Open Research Fund of Science and Technology Innovation


Platform of Engineering Technology Research Center of Dongting Lake Flood Control and Water
Resources Protection of Hunan Province, Hunan Water Resources and Hydropower Survey,
Design, Planning and Research Co., Ltd (HHPDI-KFKT-202304).
**Availability of Data and Material:** The data that support of this study are available from the
corresponding author upon reasonable request.
**Competing Interests:** We declare that we have no conflict of interest or the authors do not have
any possible conflicts of interest, the authors are not affiliated with or involved with any
organisation or entity with any financial interest or non-financial interest in the subject matter or
materials discussed in this paper.
**Ethics Approval:** Not applicable.
**Consent to Publish:** Not applicable.
**Authors Contributions:** Conceptualization, F.W. and S.P.; Methodology, X.Z.; Supervision, F.Z.;
Formal analysis, F.W.; Investigation, Y.W. and X.Z.; Writing original draft, W.W. and H.S.    And
all authors read and approved the manuscript.
**Consent to Participate:** Not applicable.
**Open Research:** The data come from the Resource and Environmental Science and Data Center
and the Henan Provincial Water Resources Bulletin. The data of 59 meteorological stations in the
Yellow River Water Supply Area in Henan were collected. The data of precipitation, temperature
and soil were based on 58 years (1961-2018) monthly data.
The datasets come from the following:
China Soil Analysis Dataset; PerMonthy and China natural Runoff
These data are ArcGIS data, is the original data, need to be processed to use.



Our data is non-public and private.
Our data is currently being archived and the planned database is Mendeley Date.

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
