# Peer review of "Study on Multi-water Sources Allocation Based on Multi-scenario potential"

_Natural Hazards and Earth System Sciences, 2024_

## Author Comment (AC1)

**General comments**

The study assessed the potential to optimise water supply allocation during extreme drought years in the Yellow River Basin. Specifically, three water supply scenarios were created to optimise different management measures in order to minimise water shortages. The water supply scenarios aimed to explore the potential for rainwater harvesting, water storage and groundwater abstraction (i.e. "unconventional" sources) to supplement surface water abstraction (i.e. "conventional" sources) during an extreme drought year (2010). The authors provided interesting insights into the measures to mitigate drought impacts and maximise water supply security, including considering sustainable and equitable water use across sectors.

However, whilst the topic is relevant to NEHSS and the wider implications of the main results could be an important contribution to improving water resources management, the paper requires a clearer structure, a more comprehensive description of the methods and a more critical discussion of the results. I would therefore recommend major revisions before this paper can be reconsidered for publication. To improve the paper, I would suggest including distinct methods and results sections to make clear which sections of the paper are results generated by the authors and which sections are information taken from secondary sources. Additionally, it is unclear from the paper how the water supply scenarios were modelled, how the water resources allocation model is parameterised and how the scenarios are applied. A number of major and additional comments (denoted by line numbers) are further presented below. I hope my comments will help the authors improve their paper.

- **Methods**:There should be a Methods section clearly detailing the definition of each scenario and exactly how these scenarios are defined within the water resource allocation model. It is hard to decipher how the various equations presented fit together-perhaps a flow chart could help here to illustrate the various inputs, outputs, model parameters and what, if any, algorithms were used to solve for the optimal water resources allocation. There is mention of the genetic algorithm

used to optimise the model solution in the Discussion section but that information should ideally be placed in a methods section and explained more fully. The authors could also consider providing details on whether the model has been tested or validated in the methods section. I would also appreciate more information on how the water supply scenarios were modelled (e.g. were the model coefficients presented varied in some ways to represent the specifications of each scenario?).

Reply: First, about the definition of the situation, before the repair, in the "2.2 potential scenario setting" part has been described, the specific definition is as follows: according to the strength of water-richness, different scenarios are tapped. From scenario 1 to scenario 3, the supply potential increases in turn, and different supply increase scenarios correspond to different supply increase measures.

Secondly, about the embodiment of water supply scenario in the model, the formula and constraint conditions of water supply in the model have been described. Thirdly, about the algorithm of multi-water source allocation, the "3.4 model solution" part is added, and the process and parameters of the solution are introduced. The specific new contents are as follows:

Genetic algorithm is a kind of intergenerational evolution, survival of the fittest, from low to high level algorithm. This algorithm takes the optimization of the global as the goal, carries out random search in the feasible solution space, realizes the group replacement and iterative optimization through cross compilation, and makes each individual gradually reach the optimal until the evolution of each generation. The genetic algorithm has strong adaptability, can independently optimize and search the solvable space, has fast convergence speed and does not depend on the decoding process, and the large search space greatly improves the calculation accuracy.

(1)The calculation steps of multi-water allocation algorithm are as follows:

The first step: Let the random scale value $A$, the initial population $B_0$, and $B_0$ as the parent population;

The second step: calculate the crowding distance and the order of the parent population, and optimize the selection, crossover and mutation to obtain the offspring population $B_t$ ;

The third step: merge $B_0$ and $B_t$ to obtain a scale of $2A$ , and the resulting offspring population is $C_t$ .After sorting, the individual crowding degree is calculated, and the previous $A$ individuals are selected to form the parent population $B_{t+1}$ ;

Step 4: judge whether the result meets the conditions, if not, go back to the second step, and output the result if it meets.

(2)The decision variables are written as follows:

This paper considers five water sources (surface water, elastic groundwater exploitation, flood resources, unconventional water) and three water supply scenarios (Scenario 1, Scenario 2, Scenario 3), four water sectors (domestic, industrial, agricultural, and ecological water). According to the constraints of the multi-water allocation model, the decision variables are numbered as follows (Table 7), $x_{i,j}$ represents the water supply of different water supply sources to different water use sectors. Among them, $i$ represents the type of water supply source, ( $i$ =1,2,3,4,5), $j$ represents different water use sectors, ( $j$ =1,2,3,4), $p$ represents different water supply scenarios, ( $p$ = 1,2,3).

Table 7 Numbering of decision variables

| Water use Department | Life | Industry | Agriculture | Zoology |
|---|---|---|---|---|
| Surface water | $x_{11,\,p}$ | $x_{12,\,p}$ | $x_{13,\,p}$ | $x_{14,\,p}$ |
| Unconventional water | $x_{21,\,p}$ | $x_{22,\,p}$ | $x_{23,\,p}$ | $x_{24,\,p}$ |
| Groundwater | $x_{31,\,p}$ | $x_{32,\,p}$ | $x_{33,\,p}$ | $x_{34,\,p}$ |
| Flood resource utilization | $x_{41,\,p}$ | $x_{42,\,p}$ | $x_{43,\,p}$ | $x_{44,\,p}$ |

(3) Function call and optimal solution selection:

The gamultiobj function needs to be called during the operation of the genetic algorithm. The function expression to be called is [ x, fval ] = gamultiobj ( fitnessfcn, nvars, A, b, Aeq, beq, lb, ub, options ), x is the pareto solution set obtained by the gamultiobj function, fval is the objective function, nvars is the total amount of variables, options is the genetic parameter, fitnessfcn is set as the handle of the objective function, which is regarded as the fitness function. A, Aeq, b and beq are the constraint conditions of the function, and ub and lb are the upper and lower limits of the constraint values. The genetic parameters of the genetic algorithm are set as follows : the individual coefficient is 0.3, the highest evolution generation is 3000, the end generation is 3000, and the fitness function error value is 0.0001.

The optimal solution is distributed in the pareto solution set, and the minimum water shortage of domestic water demand, industrial water demand, agricultural water demand and ecological water demand is comprehensively considered. Finally, the optimal solution is selected. Under the condition of satisfying the available water supply and related constraints, the priority of domestic water supply is considered, then the industrial water supply is considered, then the agricultural water supply is considered, and finally the ecological water supply is considered.

**Results:** The results from this paper seems to be Table 7, which shows the water availability for each sector and each scenario after solving the water resources allocation model. However, this section is very short with limited description of the main results. Table 8 compares the scenario results with actual water resources availability during 2010 but this is just a repetition of what is already shown in Table 7. Instead of a large table like Table 7, perhaps some figures summarising the results visually would be helpful and could replace repeated information in Table 8. The inclusion of water demand satisfaction estimation in Table 10 is interesting and seems novel but it is not explained in the paper how water demand satisfaction is estimated, how water demand is considered in each of the scenarios and how they are included in the water resources allocation model.

Reply: Table 7 gives a detailed summary of the water supply in the living, industrial, agricultural and ecological sectors of each sub-region based on the three

water supply scenarios. Therefore, a lot of space is saved, and only the water supply and water shortage under different scenarios are expressed. Table 8 analyzes the changes of water supply before and after tapping the potential from the perspective of different water supply departments, and it is also a more detailed interpretation of table 7. The water demand satisfaction in Table 10 refers to the degree of satisfaction of water demand ( the proportion of water supply in each water department to the water demand of each department ). The water demand satisfaction is calculated based on the results of multi-water source allocation, and does not need to be included in the multi-water source allocation model.

**Discussion:** This section is very short and there is very limited discussion of how the results fit with the wider literature on drought mitigation measures (such as relevant studies in nature-based solutions for drought mitigation) in the wider region and/or globally. While I appreciate that findings often differ between studies due to different methods and spatial scales, there should be much more critical discussion of how your results relate to the wider scientific literature in drought management. Additionally, there is no discussion of possible sources of uncertainties associated with the water resources allocation model. For example, the authors could consider providing some discussion of the assumptions of the model and how that may influence the reliability of the results.

Reply: First, in the discussion, a comparative discussion between the results of this study and the national scientific literature was added. The specific new contents are as follows :Firstly, Compared with the research results of other scholars (Wang, 2024), the literature (Wang, 2024) takes the Yellow River Basin as the research object, the spatial scale of the research is relatively large, and the water resources allocation scheme obtained is not specific enough. This paper takes the Henan water supply area of the Yellow River as the research object. The multi-water source water supply scheme is specific to the district and county levels, which improves the accuracy of multi-resource allocation, and more scientifically reveals the drought and supply and demand conditions at a smaller spatial scale. The multi-water source tapping potential based on multiple scenarios has greatly alleviated the drought status of the Henan

Yellow River water supply area, reduced the water supply pressure and water use restrictions of conventional water sources, and improved the support capacity and guarantee capacity of water resources. In the future, it is necessary to further optimize the research methods and improve the accuracy of data to enhance the rationality and scientificity of multi-water source allocation schemes.

Secondly, about the uncertainty problem related to the configuration model, some problems of the model algorithm are added in the ' 4.3 discussion ' section. The specific contents are as follows: Due to the randomness of crossover and mutation operations, the algorithm may fall into the local optimal solution in the search process, and the global optimal solution cannot be found, which may affect the accuracy and reliability of the solution. At the same time, the parameter settings in the genetic algorithm, such as crossover rate and mutation rate, have an important impact on the solution results. However, the setting of these parameters often needs to be adjusted according to specific problems, and there is no unified setting method at present. Different parameter settings may lead to the instability of the solution results. In the future, the algorithm needs to be further optimized.

**Additional comments:**

L38–please explain what you mean by "uneven water cycle". This does not seem to be consistent with the language used by the IPCC.

Reply: The "uneven water cycle" here has been changed into the contradiction between supply and demand of water resources. The contradiction between supply and demand of water resources refers to the imbalance between supply and demand of water resources. This contradiction is usually manifested in the supply of water resources can not meet the needs of human life and economic development, or in some areas, the development and utilization of water resources more than the carrying capacity of the local environment, resulting in the shortage of water resources or excessive consumption.

L49–this is the first time "unconventional water resources" are used in the text. "Conventional" and "unconventional" should be introduced, perhaps with examples, from the start. For example, the section from L172-182 explaining rainwater

harvesting, reclaimed water, water storage capacity and groundwater abstraction should be placed much earlier.

Reply: The concept of conventional water and unconventional water has been added in the "Introduction" section. The new content is as follows: conventional water resources (which are widely used in daily life, easy to obtain, and can be directly used for human activities after appropriate treatment, such as surface water, groundwater, tap water); unconventional water (which refers to water resources other than conventional water resources such as surface water and groundwater in the traditional sense, such as reclaimed water, rainwater, seawater, mine water, brackish water, etc.)

L56-91–instead of listing out studies one after the other, it would be more insightful if you identified common themes, methods and findings from previous studies which motivated the study aims.

Reply: We are willing to accept the proposal. In the future research, we will improve the insight of academic research, find problems such as themes and methods from the research, and carry out targeted and insightful research.

L129–"drought change of the PDSI annual sequence"–do you mean drought occurrence?

Reply: "drought change of the PDSI annual sequence" is based on the PDSI value to determine whether the drought exists and the degree of drought, according to the size of the PDSI value to divide the drought level (as shown in table 2).

Table 2 PDSI drought classification standard table

| PDSI | Drought level | PDSI | Drought level |
|------|---------------|------|---------------|
| (-1,1) | Normal | (-4,-3] | Serious drought |
| (-2,-1] | Light drought | (-∞,-4] | Extreme drought |
| (-3,-2] | Moderate drought | | |

L138–"every 10 or so"–is this referring to years?

Reply: "every 10 or so" refers to the probability of severe drought (on average, one severe drought every 10 years).

L140–drought severity rather than "grade" might be clearer.

Reply: The drought severity is expressed according to the drought level. The drought level can be divided into no drought, light drought, moderate drought, severe drought and extreme drought. Therefore, the drought level is used to describe the severity of drought.

L144-145–repetition of L139

Reply: "the drought degree of the Yellow River Water Supply Area in Henan reached the level of extreme drought in 2010." has been deleted from the original text.

L183–What do you mean by "water richness" and how is it quantified?

Reply: "water richness" refers to the water yield capacity of aquifer, which is a sign to measure the water yield of aquifer during groundwater exploitation. In this paper, the water-rich grade of groundwater in each zone is judged according to the unit water output of the water source well and the spring water flow. The specific quantitative division standard is referred to Table 4.

Table 4    Division table of groundwater water abundance in
the Yellow River Water Supply Area in Henan

| Regionalization basis | Partition | | | |
|---|---|---|---|---|
| | Weak water-rich area | Medium water-rich area | Strong water-rich area | Extremely strong water-rich area |
| Unit output of water source well(m³/h·m) | $q<1$ | $1\leq q<5$ | $5\leq q<10$ | $q>10$ |
| Flow capacity of spring (L/s) | $Q<1$ | $1\leq Q<10$ | $10\leq Q<5$ | $Q<50$ |

L183–As noted in general comments, it is not clear from the text what "tapping" means–do you mean the different scenarios are tested adopted depending on groundwater abundance of the region? Both "tapping" and "digging" potential are used throughout the text but neither terms are properly defined–are they different concepts or do they refer to the same thing?

Reply: "tapping" means that under extreme drought conditions, different degrees of potential tapping are carried out according to different supply increase measures (see Table 5). From water supply scenario 1 to water supply scenario 3, the degree of potential tapping gradually increases. "tapping" and "digging" express the same thing, that is, the meaning of digging, and "digging" has been completely changed to "tapping".

Table 5 Measures for increasing supply of different potential water sources under different supply scenarios

| Additional supply scenario | Unconventional Water Tapping | Flood resource utilization | Elastic groundwater exploitation |
|---|---|---|---|
| Scenario 1 | Increase the scale of rainwater harvesting facilities by 5% | Increase the scale of water storage project by 5% | Mining 15% of water source in strong water-rich area and extremely strong water-rich area |
| Scenario 2 | Expand the scale of 5% rainwater harvesting facilities; reclaimed water and sewage utilization efficiency increased by 10% | Increase the scale of water storage project by 5%; reasonable setting to speed up the recovery scheduling rules | Mining 15% of the water source in the strong and extremely rich water area; mining and excavating 10% of the water source in the medium water-rich area |
| Scenario 3 | Expand the scale of rainwater harvesting facilities by 5%;the utilization efficiency of reclaimed water and sewage is increased by 10%;increase the utilization ratio of mine water by 20% | Increase the scale of water storage project by 5%; reasonably set up the scheduling rules for accelerating the recovery of savings; dynamic adjustment of reservoir flood control level | Mining 15% of the water source in the strong and extremely rich water area; mining and excavating 10% of the water source in the medium-rich water area; mining 5% water source in weak water-rich area |

Table 4–What does q and Q stand for in the table? The relevance of this table is not clear and it is unclear how it relates to the water supply scenarios listed in Table 5.

Reply: In Table 4, $q$ and $Q$ represent the unit water yield of water source wells and spring water flow respectively. According to these two indicators, they are only used to judge the water abundance of groundwater in each partition and carry out the elastic tapping of groundwater. Table 5 is from the perspective of percentage, based on unconventional water, flood resources, groundwater potential water, to tap the potential of different scenarios.

Table 6–Does this table report results obtained from the study or are they values taken from secondary sources.

Reply: The unconventional water, flood resource utilization and groundwater volume in Table 6 are calculated according to the potential water volume. The potential water volume has been explained in the part of "3.1 Water Source Analysis of Multi-Scenario Potential Exploitation" (i.e. the potential water volume of unconventional water in Henan Yellow River Water Supply Area in 2020 is 50.45 billion m3, the potential water volume of flood resource utilization is 10.223 billion m3, and the potential water volume of groundwater elastic exploitation is 9.660 billion m3). At the same time, the potential water volume of different scenarios is calculated according to the percentage and the potential water volume of various water sources. The percentage is based on the comprehensive setting of the supply capacity and scale of rainwater harvesting facilities, water storage projects, etc.

Section 3.2–Did the authors come up with the water demand hierarchy themselves? If not, there should be appropriate reference to previous studies which have applied similar concepts.

Reply: In the original text, the "3.2 Water demand level division and configuration principle" section has put forward the level of water demand, which divides the water demand process of life, industry, agriculture and ecology into three levels, namely rigid demand, rigid elastic demand and elastic demand. Rigid water demand is in the first priority in the water distribution, and once it is destroyed, it will be difficult to recover the loss; rigid elastic water demand is the second priority in the distribution water, and the loss caused by water shortage is recoverable; elastic water demand is the last consideration in water resources allocation.

L264–what does "enemy" mean?

Reply: Has been corrected to "Where $Z_p$ is the total water shortage of scenario $p$"

Open research section: are these meant to be hyperlinks to the data source? If so, the links don't seem to be working.

Reply: The data in this paper are derived from the paper-based statistical yearbook of Henan Province over the years, and cannot generate hyperlinks. At the same time, these data are confidential and are not provided, please understand. However, in the "1.2 Basic Information" section, references to data sources have been added. The new literature is as follows:

[27] Kang J F, Zhang Y N, Liu C, et al. Human Economic Data Set of the Yellow River Basin from 2015 to 2019, China Scientific Data, 7(04): 118-132, 2022.

[28] Li J, Nie H M, Xu G Z. Characteristics Analysis on Carbon Reduction of Crop Production in Henan Province Based on the Statistical Yearbook Data, Chinese Journal of Agrometeorology, 44(09): 759-768, 2023.

[29] Zheng Z. Regional Water Consumption Characteristics and Trend Forecast under the Most Stringent Water Resources Management System, North China University of Water Resources and Electric Power, 2018.

---

## Author Comment (AC2)

**Author's response**

Wan and co-authors developed a multi-source water allocation model to evaluate the integration of alternative water supply sources under multiple drought scenarios. They developed and implemented this model for the Yellow River in the Hennan province of China. The topic is a good fit for NHESS but the manuscript needs considerable improvement before being suitable for publication. Key areas for improvement include description and citation of the data used, additional details on the model development and configuration, and clear concise writing throughout. In revising the paper it might be helpful to think of a student or early career researcher looking to apply similar methods to their project as the reader.

**Comments**

1.Little information is provided about the data used in this work. What soil variables were used? Are there any data gaps in the time series data for temperature and precipitation? If so, what percent of the data is missing and how were gaps filled? Additionally, please cite the sources for each variable in the data or set of variables used in this work. A table would be an efficient way to present this information. See for example Table 1 in Garcia and Islam (2021).

Re: The data used in this paper is the Palmer Drought Index (PDSI), which is used to characterize the evolution of drought in the Yellow River water supply area of Henan Province. The precipitation, temperature and soil data are used to calculate the Palmer Drought Index. In order to improve the integrity of the data, the data type and its source description are added in the "1.2 basic data" section of the text (Table 2). The literature corresponding to Garcia and Islam has been cited.

Table 2 Data Types and Data Sources Statistics

| Data type | Data scale | Data sources | Data unit |
|---|---|---|---|
| Precipitation | Month by month | Institute of Geographical Sciences and Natural Resources Research, CAS; Geographical Information Monitoring Cloud Platform | mm |

| Temperature | Month by month | Institute of Geographical Sciences and Natural Resources Research, CAS; Geographical Information Monitoring Cloud Platform | °C |
|---|---|---|---|
| Soil humidity | Month by month | Institute of Geographical Sciences and Natural Resources Research, CAS; Geographical Information Monitoring Cloud Platform | % |

2.Figure 2 is labeled a histogram but is not in fact a histogram. Histograms have the variable magnitude on the x axis and the frequency on the y axis. It is a diagram consisting of bars of even width whose height is proportional to the frequency of a variable. Figure 2 is a time series bar graph. Please see Hesel et al. (2020) for additional guidance on data visualization.

Re: Figure 2 is a histogram that reflects the overall PDSI sequence changes in the water supply area of the Yellow River in Henan Province. The x-axis represents the year, and the y-axis represents the PDSI value. This diagram has drawn on the histogram in the literature Hesel et al (2020). However, because the y value of the histogram of PDSI sequence changes is drawn according to the boundary of 0, greater than 0 and less than 0 represent wet and dry states respectively, which improves the degree of visualization and more clearly reflects the change state of dry and wet alternation.

3.The scenarios developed to augment water supplies during drought years include use of harvested rainwater, wastewater recovery, flood water recovery and groundwater. As described in Table 6, these scenarios assume that a specific amount of water will be available from these water sources will be available during those drought years. How has this been determined? For example, expanding the reservoir and changing the operating rules does increase the probability of carrying over water from high flow years to extreme drought years but it depends on the sequence of flows observed. What analysis was conducted to establish that this volume could be stored? With what probability will it be available? Similarly, precipitation will be lower during extreme drought conditions so how reliable will the harvested rainwater be? What storage capacity and use rules are needed to have this amount of water available from rainwater harvesting with high reliability?

Lastly, depending on the local hydro-geological conditions, groundwater levels and flow rates may decline during drought. Are all aquifers in this watershed unaffected by drought?

Re: First, the unconventional water, flood resources, and groundwater volume in Table 6 are calculated based on the potential water volume. The potential water volume has been explained in the section of the original "3.1 Multi-scenario Potential Water Source Analysis" (that is, the potential water volume of unconventional water in the Henan Yellow River water supply area in 2020 is 5.045 billion m$^3$, the potential water volume of flood resources is 10.223 billion m$^3$, and the potential water volume of groundwater elastic mining is 9.660 billion m$^3$. These potential water volumes are not used or rarely used in non-arid years),at the same time, the potential water volume of different scenarios is calculated according to the percentage and the potential water volume of various water sources, and the percentage is based on the comprehensive setting of the supply capacity and scale of rainwater collection facilities, water storage projects, etc.

Secondly, in the case of extreme drought, the spatial and temporal distribution of precipitation is uneven, and the scale of rainwater collection and storage can be expanded in the concentrated rainy season and strong precipitation areas. The construction of facilities needs to consider the local rainfall characteristics, topography and soil conditions, while minimizing the use of multi-functional storage facilities to ensure sufficient water storage capacity during drought periods; improve various rainwater collection systems, such as source interception measures, initial rainwater abandonment or diversion, which helps to reduce pollutants in rainwater and improve the quality of rainwater collection.

Third, during the drought period, the supply of rainfall and surface water will be reduced, resulting in a decrease in the supply of groundwater, which will affect the amount of water in the aquifer. The elastic mining of groundwater in this paper is based on extreme drought. Considering the groundwater enrichment and potential water volume of each partition, under the premise of ensuring the sustainable utilization of groundwater resources, scientific and reasonable mining is carried out. Or under extreme drought, there will be over-exploitation of groundwater, and groundwater will be recharged during the wet year.

4.It is not clear how the model developed addresses the spatial variation in water demands and supplies. Can these alternate water sources be developed to the same degree in all regions? Is

sufficient water available in the correct locations to fully meet demands? If not, what infrastructure assumptions are made?

Re: The multi-water source model is based on the consideration of available water supply constraints, water demand levels and configuration principles, and proposes a fair, scientific and reasonable configuration plan for the county level to alleviate the drought situation in the Yellow River water supply area of Henan to the greatest extent. Due to the difference of groundwater enrichment and reservoir structure distribution, the development degree of each partition in Henan Yellow River water supply area is different. The potential tapping based on different water supply scenarios cannot fully meet the water demand of Henan Yellow River water supply area. This paper considers the multi-water source configuration in extreme drought years, which can alleviate the drought status of Henan Yellow River water supply area to the greatest extent. This paper has made potential tapping measures in different scenarios (Table 5), and has not proposed infrastructure assumptions from the perspective of implementation.

Table 5 Measures for increasing supply of different potential water sources under different supply scenarios

| Additional supply scenario | Unconventional Water Tapping | Flood resource utilization | Elastic groundwater exploitation |
| --- | --- | --- | --- |
| Scenario 1 | Increase the scale of rainwater harvesting facilities by 5% | Increase the scale of water storage project by 5% | Mining 15% of water source in strong water-rich area and extremely strong water-rich area |
| Scenario 2 | Expand the scale of 5% rainwater harvesting facilities; reclaimed water and sewage utilization efficiency increased by 10% | Increase the scale of water storage project by 5%; reasonable setting to speed up the recovery scheduling rules | Mining 15% of the water source in the strong and extremely rich water area; mining and excavating 10% of the water source in the medium water-rich area |
| Scenario 3 | Expand the scale of rainwater harvesting facilities by | Increase the scale of water storage project by 5%; | Mining 15% of the water source in the strong and extremely rich water |

| 5%;the utilization efficiency of reclaimed water and sewage is increased by 10%;increase the utilization ratio of mine water by 20% | reasonably set up the scheduling rules for accelerating the recovery of savings; dynamic adjustment of reservoir flood control level | area; mining and excavating 10% of the water source in the medium-rich water area; mining 5% water source in weak water-rich area |

5.The methods description leave me with many questions about the model is set up and run. Have the authors used streamflow from only 2010 for the optimization? How were initial conditions such as reservoir levels determined where considering reservoir expansions and changing operating rules? Is demand assumed to be constant at 2010 levels? Is the model fully deterministic or are there stochastic elements? What software or programming language was used to implement the model? What algorithm was used for optimization?

Re: In this paper, 2010 is selected as the extreme drought year, and 2010 is used as the demand level of multi-water source allocation. The initial conditions are determined based on historical data, hydrological and meteorological conditions, reservoir design standards, etc. The multi-water source allocation model is not completely determined, and there may be uncertainties in parameters and solving process. The model in this paper is solved by genetic algorithm, and the population replacement and iterative optimization are optimized by cross-compilation. The individual gradually reaches the optimal until. In this paper, the "3.4 model solution" section is added, and the new contents are as follows:

Genetic algorithm is a kind of intergenerational evolution, survival of the fittest, from low to high level algorithm. This algorithm takes the optimization of the global as the goal, carries out random search in the feasible solution space, realizes the group replacement and iterative optimization through cross compilation, and makes each individual gradually reach the optimal until the evolution of each generation. The genetic algorithm has strong adaptability, can independently optimize and search the solvable space, has fast convergence speed and does not depend on the decoding process, and the large search space greatly improves the calculation accuracy.

(1) The calculation steps of multi-water allocation algorithm are as follows:

The first step: Let the random scale value $A$, the initial population $B_0$, and $B_0$ as the parent population;

The second step: calculate the crowding distance and the order of the parent population, and optimize the selection, crossover and mutation to obtain the offspring population $B_t$;

The third step: merge $B_0$ and $B_t$ to obtain a scale of $2A$, and the resulting offspring population is $C_t$. After sorting, the individual crowding degree is calculated, and the previous $A$ individuals are selected to form the parent population $B_{t+1}$;

Step 4: judge whether the result meets the conditions, if not, go back to the second step, and output the result if it meets.

(2) The decision variables are written as follows:

This paper considers five water sources (surface water, elastic groundwater exploitation, flood resources, unconventional water) and three water supply scenarios (Scenario 1, Scenario 2, Scenario 3), four water sectors (domestic, industrial, agricultural, and ecological water). According to the constraints of the multi-water allocation model, the decision variables are numbered as follows (Table 7), $x_{i,j}$ represents the water supply of different water supply sources to different water use sectors. Among them, $i$ represents the type of water supply source, ($i$ =1,2,3,4,5), $j$ represents different water use sectors, ($j$ =1,2,3,4), $p$ represents different water supply scenarios, ($p$ = 1,2,3).

Table 7 Numbering of decision variables

| Water use sector | Life | Industry | Agriculture | Zoology |
|---|---|---|---|---|
| Surface water | $x_{11,\ p}$ | $x_{12,\ p}$ | $x_{13,\ p}$ | $x_{14,\ p}$ |
| Unconventional water | $x_{21,\ p}$ | $x_{22,\ p}$ | $x_{23,\ p}$ | $x_{24,\ p}$ |
| Groundwater | $x_{31,\ p}$ | $x_{32,\ p}$ | $x_{33,\ p}$ | $x_{34,\ p}$ |
| Flood resource utilization | $x_{41,\ p}$ | $x_{42,\ p}$ | $x_{43,\ p}$ | $x_{44,\ p}$ |

(3) Function call and optimal solution selection:

The gamultiobj function needs to be called during the operation of the genetic algorithm. The function expression to be called is [x, fval] = gamultiobj (fitnessfcn, nvars, A, b, Aeq, beq, lb, ub, options), x is the pareto solution set obtained by the gamultiobj function, fval is the objective function, nvars is the total amount of variables, options is the genetic parameter, fitnessfcn is set as the handle of the objective function, which is regarded as the fitness function. A, Aeq, b and beq are the constraint conditions of the function, and ub and lb are the upper and lower limits of the constraint values. The genetic parameters of the genetic algorithm are set as follows : the individual coefficient is 0.3, the highest evolution generation is 3000, the end generation is 3000, and the fitness function error value is 0.0001.

The optimal solution is distributed in the pareto solution set, and the minimum water shortage of domestic water demand, industrial water demand, agricultural water demand and ecological water demand is comprehensively considered. Finally, the optimal solution is selected. Under the condition of satisfying the available water supply and related constraints, the priority of domestic water supply is considered, then the industrial water supply is considered, then the agricultural water supply is considered, and finally the ecological water supply is considered.

**Minor Comments**

1.The sentence starting on line 40 ("In recent years…") is not a complete sen

Re: "In recent years..." has been changed to "The Yellow River Water Supply Area in Henan with monsoon climate characteristics has been dry and wet imbalance in recent years".

2.The use of the word "staged" in describing drought occurrence (see line 43 for example) is not appropriate because to stage means to produce or arrange and it implies human control while drought is a (mostly) natural phenomena. You could use the verb occurred in place of staged.

Re: "staged" has been changed to "occurred".

3.What is meant by the "social economy" on line 44? Is this different that the economy?

Re: Social economy is a concrete and systematic concept, which refers to the overall system formed on the basis of social structure and economic structure, through the condensation of material resources and social relations, and focuses on the economic behavior of human beings in

the process of social movement. Economy is a broader concept, which refers to the production, circulation, distribution and consumption of all material and spiritual materials. Its core is the creation, transformation and realization of value to meet the needs of human material and cultural life. For social and economic development, water resources are indispensable basic support conditions. Daily life, industrial production or agricultural production all need a lot of water resources to meet the demand.

4.What is the elastic exploitation of groundwater? In particular, what does elastic mean here? Please clarify for the reader.

Re: Elastic exploitation of groundwater is a strategy for groundwater exploitation and management. It allows the dynamic adjustment and exploitation of groundwater resources within a certain range according to the recharge capacity and hydrogeological conditions of the groundwater system under appropriate conditions. The meaning of the word "elasticity" refers to the ability of the groundwater system to maintain self-regulation and recovery within a certain range in the face of mining pressure. This ability allows the groundwater system to make corresponding water level and water volume adjustments according to changes in the amount of exploitation within a certain time scale, without causing system collapse or irreversible damage.

5.The sentence starting on line 58 continues until line 74. It's hard to follow. Please consider breaking this up into multiple sentences.

Re: This part has been modified. "Wang Yu et al. (2021) scientifically set the water diversion index in the river according to the incoming water situation, and at the same time consider fairness and efficiency, and increase the saved water supply to the provinces along the Yellow River; Yang Mingzhi et al. (2022) regarded social water use and natural hydrology as the research object, studied the feedback between the two processes, and developed a distributed allocation model based on the water cycle; Tan et al. (2018) took the unilateral water benefit as the objective function, considered the fractional programming and robust optimization at the same time, and established the water resources optimization model, which improved the utilization efficiency of agricultural water; Ren et al. (2017) gave full play to the advantages of multi-objective fuzzy programming, constructed a multi-objective model of multiple benefits, rationally planned land use and irrigation water, and obtained an effective and fair irrigation plan; Aiming at the prediction

of water supply and demand and its comprehensive value, Zhang et al. (2023) used the WRA model to study the coordination and stable development of each system, used the emergy analysis method to carry out quantitative analysis, reasonably calculated the base year and the planning year, and proposed a sustainable water distribution plan; Sperotto A et al. (2019) discussed the application of multi-scenario analysis method based on Bayesian network in water quality sustainability assessment under uncertain conditions." has been changed to"Wang Yu et al. (2021) set the water diversion index in the river according to the science of the water situation, and consider the fairness and efficiency at the same time, and increase the saved water supply to the provinces and regions along the Yellow River; Yang Mingzhi et al. (2022) took social water use and natural hydrology as the research object, studied the feedback between social water use and natural hydrology, and developed a distributed allocation model based on water cycle; Tan et al. (2018) taking the single water benefit as the objective function, considering the fractional programming and robust optimization, the water resources optimization model is established to improve the utilization efficiency of agricultural water; Ren et al. (2017) giving full play to the advantages of multi-objective fuzzy programming, a multi-objective model of multiple benefits is constructed to rationally plan land use and irrigation water, and an effective and fair irrigation scheme is obtained; Zhang et al. (2023) aiming at the prediction of water supply and demand and its comprehensive value, the WRA model is used to study its coordinated and stable development, and the emergy analysis method is used for quantitative analysis. The base year and the planning year are reasonably calculated, and the sustainable water distribution scheme is proposed; Under uncertain conditions, Sperotto A et al. (2019) discussed the application of multi-scenario analysis method based on Bayesian network in water quality sustainability assessment"

6.Many acronyms are introduced in the introduction and not later used such as MS-MPC, UDN, etc. Please only introduce an acronym if it will be repeatedly used.

Re: MS-MPC, UDN, IPCC, ROs, GA, NSGA-II, MPC (MS-MPC) appeared only once in the text, and did not appear again. In order to facilitate readers understanding, the above abbreviations have been changed to "Genetic Algorithm (GA), Non-dominated Sorting Genetic Algorithm II (NSGA-II), Model Predictive Control (MPC), Multi-scenario MPC (MS-MPC), Urban Drainage Networks (UDN), Real Options (ROs)".

7.What is the definition of "tapping potential"? I am not familiar with this term and do not think it is commonly used in water resource engineering.

Re: "tapping potential of water resources" is a new vocabulary in the discipline of water resources. "tapping potential" refers to the excavation, development and exploration of potential capabilities or resources. Therefore, "tapping potential" refers to the excavation and utilization of certain potential capabilities or resources through certain methods and measures. Specifically, "tapping potential of water resources" refers to excavating and utilizing the unutilized or potential parts of water resources (such as unconventional water and flood resources) through scientific planning, technological innovation and management improvement, so as to increase the effective supply and rational utilization of water resources and alleviate the contradiction of water shortage.

8.What does mu refer to on line 106 and 107? It looks like a unit, but I am not familiar with this unit or its abbreviation.

Re: "mu" refers to the unit of land area, one mu is equal to 666.7 square meters, and the unit in English is "mu".

9.What does it mean for precipitation to be unbalanced spatially and temporally? (As on lines 108-109)? Do you mean that there is high spatial and temporal variability?

Re: The imbalance of precipitation in time and space means that the distribution of precipitation is uneven in time and space. For example, in the monsoon climate zone, summer precipitation is concentrated, while winter precipitation is scarce. This seasonal uneven precipitation may lead to serious floods and drought disasters. At the same time, the uneven distribution of precipitation in space may lead to floods in some areas due to excessive precipitation, while other areas face drought and water shortage due to too little precipitation. The uneven spatial and temporal distribution of precipitation means that there are different degrees of spatial and temporal variability. The variability of time and space makes the supply of water resources in different regions at different times may also change significantly. This variability has an important impact on agricultural production, water resources management, disaster prevention and control.

10.There are a number of instances where spaces between words are missing. For example, after the word "areas" on line 112.

Re: "areas (Sun, 2021)." has been changed to "areas (Sun, 2021)."; "Index(PDSI)" has been changed to "Index (PDSI)"; "zoning(Zhao and Zhao, 2014)." has been changed to "zoning (Zhao and Zhao, 2014)."; "Ministry of Water Resources(under construction)(2023-SYSJJ-05)" has been changed to "Ministry of Water Resources (under construction) (2023-SYSJJ-05)"; "Resources(KYFB202307260036)" has been changed to "Resources (KYFB202307260036)".

11.I am curious about the increase in ecological water use mentioned on line 113. What has driven the increase in ecological water use? Is this a legal or regulatory requirement?

Re: There are many reasons for the increase in ecological water use, including the improvement of environmental awareness, the need for ecological system maintenance, the decline in the proportion of industrial and agricultural water use, engineering construction and planning, and the implementation of ecological restoration projects. These factors work together to promote the increase of ecological water consumption. With the improvement of environmental awareness and the popularization of the concept of sustainable development, many countries and regions have formulated relevant laws and policies to protect and restore ecosystems. These laws and policies may require the consideration of ecological water demand in water resources management and distribution, thus indirectly promoting the increase of ecological water use.

12.Table 1 is not needed as knowledge of these subdivisions is not needed by the reader to interpret the results.

Re: Table 1 details the scope of the Yellow River water supply area in Henan.At the same time, not all districts and counties in the municipal area are in the Yellow River water supply area in Henan.Some districts and counties in the municipal area are not in the Yellow River water supply area in Henan.Therefore, table 1 is to more accurately explain the scope of the study area.

13.Line 139 refers to mild and normal drought years. These terms are not the ones defined in Table 2. For clarity, please use the terms defined.

Re: "Mild drought" has been changed to "Moderate and slight drought".

14.In Figure 3, outline each of the geographic areas. In the current figure some colors (drought levels) are outlined while others are not.

Re: Figure 3 has been modified, and the contours of each region have been added, as shown in the following figure.

[Figure]

Figure 3    Distribution of drought grade in the Yellow River Water Supply Area in Henan in 2010

15.What does water-rich zoning refer to on line 183? This is not a common term so please explain to the reader.

Re: water-rich zoning refers to the water yield capacity of aquifer, which is a sign to measure the water yield of aquifer during groundwater exploitation. In this paper, the water-rich grade of groundwater in each zone is judged according to the unit water output of the water source well and the spring water flow. The specific quantitative division standard is referred to Table 4.

Table 4    Division table of groundwater water abundance in

the Yellow River Water Supply Area in Henan

| Regionalization basis | Partition | | | |
|---|---|---|---|---|
| | Weak water-rich area | Medium water-rich area | Strong water-rich area | Extremely strong water-rich area |

| Unit output of water source well(m³/h·m) | $q<1$ | $1\leq q<5$ | $5\leq q<10$ | $q>10$ |
|---|---|---|---|---|
| Flow capacity of spring (L/s) | $Q<1$ | $1\leq Q<10$ | $10\leq Q<50$ | $Q<50$ |

16.Do both criteria in Table 4 need to be satisfied for the classification? How are cases where the two criteria are not aligned dealt with?

Re: The classification of water abundance needs to meet the two standards of unit water yield of water source well and spring water flow at the same time. Any index does not meet the corresponding numerical range, it will not be divided into corresponding partitions.

17.In Table 6, what does "digging potential" mean? Is this the baseline scenario? Also in Table 6, all scenarios are labeled Scenario 1.

Re: "potential tapping" generally refers to excavating and utilizing certain potential capabilities or resources through various technical and engineering means to meet the needs of human production, life and ecology. Specifically, "tapping the potential of water resources" refers to excavating and utilizing the unutilized or potential parts of water resources (such as rainwater harvesting, flood recycling, seawater desalination, etc.) through scientific planning, technological innovation and management improvement, so as to increase the effective supply and rational utilization of water resources and alleviate the contradiction of water shortage. About the definition of the scenario, before the repair, in the "2.2 potential scenario setting" part has been described, the specific definition is as follows : According to the strength of water richness, different scenarios are tapped. Scenarios 1 to 3, the potential of increasing supply increases in turn, and different scenarios correspond to different measures of increasing supply.

18.In the Zp definition on line 264, who or what is the enemy?

Re: $Z_p$ is the total water deficit of scenario $p$ , " $Z_p$ is the total water deficit in the scenario of enemy $p$ „ has been replaced by " $Z_p$ is the total water shortage of scenario $p$ „.

19.There is no i in equation 1. What does variable i on line 265 refer to?

Re: There is an error in spelling, "$W_{c,p}$ is the water supply of type $i$ user in scenario $p$" has been changed to "$W_{c,p}$ is the water supply of type $c$ user in scenario $p$".

20.Can the variables introduced in equations 2 through 5 be labeled in a more intuitive way? For example, $W_G$ is total water while $W_B$ is groundwater and $W_T$ is ecological water. If $W_G$ were groundwater, $W_T$ total water and so forth, the variable definitions would be easier to remember.

Re: "$W_B$" has been changed to "$W_S$", which indicates the available water supply of surface water ; "$W_D$" has been changed to "$W_G$", which indicates the amount of groundwater available; "$W_S$" has been changed to "$W_L$", which indicates the water demand for living ; "$W_Y$" has been changed to "$W_I$", indicating industrial water demand ; "$W_N$" has been changed to "$W_A$", indicating agricultural water demand ; "$W_T$" has been changed to "$W_E$", indicating ecological water demand.

21.On line 300, should $D_0$ read $D_e$?

Re: "$D_0$" has been changed to "$D_e$".

References

Garcia, M., & Islam, S. (2021). Water stress & water salience: implications for water supply planning. Hydrological Sciences Journal, 66(6), 919-934.

Helsel, D.R., Hirsch, R.M., Ryberg, K.R., Archfield, S.A., and Gilroy, E.J., (2020). Chapter 2: Graphical Analysis. Statistical methods in water resources: U.S. Geological Survey Techniques and Methods, book 4, chap. A3, 458 p., https://doi.org/10.3133/tm4a3.